# Measurement of the Depth of Lesions on Proximal Surfaces with SWIR Multispectral Transillumination and Reflectance Imaging

**DOI:** 10.3390/diagnostics12030597

**Published:** 2022-02-26

**Authors:** Yihua Zhu, Daniel Fried

**Affiliations:** Department of Preventive and Restorative Dental Sciences, University of California, San Francisco, 707 Parnassus Ave., San Francisco, CA 94143, USA; yihua.zhu@ucsf.edu

**Keywords:** SWIR imaging, caries detection, reflectance, transillumination

## Abstract

The aim of this study was to compare the diagnostic performance of dual short-wavelength infrared (SWIR) transillumination and reflectance multispectral imaging devices for imaging interproximal lesions with radiography using extracted teeth that had been imaged with micro-computed tomography (microCT). Thirty-six extracted teeth with 67 lesions on the proximal surfaces were imaged using a newly fabricated SWIR multispectral proximal transillumination and reflectance imaging device along with an existing SWIR multispectral occlusal transillumination and reflectance device. The ability of SWIR imaging and radiography to detect lesions and accurately assess lesion dimensions were compared using microCT as a standard. Occlusal and proximal transillumination and occlusal reflectance performed best for imaging interproximal lesions while proximal reflectance was not useful for imaging interproximal lesions from tooth buccal and lingual surfaces. There was high correlation of the lesion dimensions measured in occlusal and proximal transillumination images compared to microCT and moderate correlation in occlusal reflectance images. The correlation between the lesion depth measured in radiographs and the lesion depth measured with microCT was not significant. This study demonstrates that SWIR occlusal and proximal transillumination and SWIR occlusal reflectance images are useful for imaging interproximal lesions and they provide more accurate measurements of lesion severity.

## 1. Introduction

Short wavelength infrared (SWIR) and near-IR imaging (NIR) methods have been under development for almost 20 years for use in dentistry and several NIR clinical devices are now available commercially. Due to the high transparency of enamel at longer wavelengths, multiple imaging configurations are feasible, caries lesions can be imaged using transillumination and reflectance from tooth occlusal, buccal and lingual surfaces [1,2]. In proximal transillumination, the light source and detector are placed on the buccal and lingual sides of the tooth (see Figure 1). The positions can be alternated to get images of each surface since this method will have greater sensitivity for those lesions located closer to the detector. This is the same imaging geometry used to acquire bitewing radiographs. Interproximal lesions, the lesions located at the proximal contact points in between teeth, can be imaged via all three imaging geometries. Transillumination of the proximal contact points between teeth can also be accomplished via occlusal transillumination by directing SWIR light below the crown while imaging the occlusal surface [2,3,4] (see Figure 1). The latter approach is capable of imaging occlusal lesions as well with high contrast [1,2,5,6,7,8]. In 2010, it was demonstrated that interproximal lesions that appeared on radiographs could be detected in vivo using proximal and occlusal transillumination imaging at 1310 nm with similar sensitivity [2]. This was the first step in demonstrating the clinical potential of NIR and SWIR imaging for interproximal caries detection. A later clinical study employing SWIR transillumination and SWIR reflectance imaging probes to screen premolar teeth that were scheduled for extraction established that the sensitivity of SWIR imaging was significantly higher (*p* < 0.05) than radiographs for both occlusal and interproximal lesions [9]. The sensitivity of each individual SWIR method was either individually equal to or higher than radiography.

Several shorter wavelength NIR devices that utilize either reflectance or transillumination imaging have been introduced commercially operating at 830 and 780 nm [10,11,12,13,14]. The use of shorter wavelength 830 nm NIR light was first investigated almost 20 years ago [3]. Shorter wavelengths allow the use of less expensive silicon-based detectors. However, longer wavelength SWIR light has significant advantages. The contrast of simulated lesions in enamel at 830 nm was significantly lower than at 1300 nm and they could not be imaged through the maximum thickness of enamel [3]. It is also important to point out that stains interfere significantly at wavelengths less than 1100 nm [15] and the contrast between sound and demineralized enamel is markedly higher at wavelengths beyond 1400 nm in reflectance measurements [15,16,17].

Several studies over the past decade have indicated that transillumination performs best at 1300 nm where the transparency of enamel is highest while the contrast of lesions on tooth surfaces imaged using reflectance continues to increase with increasing wavelength and is highest at 1950 nm [15,16,17]. Several studies have investigated multispectral measurements using either one modality or combining transillumination and reflectance measurements. Zakian et al. [18,19] used multiple wavelengths of SWIR hyperspectral reflectance images to estimate the severity of occlusal lesions. Since multispectral SWIR reflectance and transillumination experiments have demonstrated that the tooth appears darker at wavelengths coincident with increased water absorption, multispectral images can be used to produce increased contrast between different tooth structures such as sound enamel and dentin, dental decay, and composite restorative materials [19,20,21]. Combining measurements from different SWIR imaging wavelengths and comparing them with concurrent measurements acquired by complementary imaging modalities should provide an improved assessment of lesion depth and severity. Radiographs markedly underestimate the depth and severity of interproximal lesions and clinicians are forced to assume that lesions penetrate much deeper than indicated in radiographs [22,23,24]. If lesions are non-cavitated and limited in penetration to only the enamel they can often be arrested by chemical intervention without the need to remove the decay with the drill. Therefore, a more accurate imaging system for assessing the depth of penetration of interproximal lesions would be a significant step forward and enable clinicians to make more informed decisions regarding treatment. Moreover, a combined SWIR reflectance and transillumination clinical probe will likely reduce false positives since it is improbable that confounding structural features or specular reflection will be present in both reflectance and transillumination images. Reflectance and occlusal transillumination probes have been combined into a single probe since light is collected from tooth occlusal surfaces for both methods. Different illumination wavelengths have been used optimized for each imaging mode, namely SWIR wavelengths greater than 1400 nm for reflectance and 1300 nm for transillumination. Simon et al. [25] built a benchtop simultaneous SWIR reflectance and transillumination system with tunable filters that ranged from 830–1700 nm and showed that the combined images have potential for the diagnosis of occlusal lesions [26] and simulated cavitated and non-cavitated interproximal lesions [27]. A compact system that is capable of simultaneous acquisition of SWIR reflectance and occlusal transillumination images suitable for clinical use was developed and tested on 120 teeth extracted teeth with occlusal and interproximal lesions using microCT as a gold standard [28,29,30]. That system is currently being used for clinical studies. The purpose of this study is to develop a second dual SWIR transillumination and reflectance probe designed to image from tooth buccal and lingual surfaces and assess the performance of this new dual probe and the existing occlusal transillumination and reflectance probe for imaging interproximal lesions on extracted teeth.

## 2. Materials and Methods

### 2.1. Sample Preparation

Teeth with no identifiers were collected from patients in the San Francisco Bay area and Geneva Switzerland with approval from the UCSF Committee on Human Research. Extracted teeth (n = 36) were selected with 67 interproximal lesions for this study. Teeth were sterilized using gamma radiation and stored in 0.1% thymol solution to maintain tissue hydration and prevent bacterial growth. Then, samples were mounted in black orthodontic acrylic blocks from Great Lakes Orthodontics (Tonawanda, NY, USA) and imaged with digital radiographs using a CareStream 2200 System from Kodak (Rochester, NY, USA) operating at 60 kV.

All teeth were imaged using Microcomputed X-ray tomography (µCT) with a 10-µm resolution. A Scanco µCT 50 from Scanco USA (Wayne, PA, USA) located at the UCSF Bone Imaging Core Facility was used to acquire the images. Visible color images of the samples were acquired using a USB microscope, Model AM7915MZT from AnMO Electronics Corp. (New Taipei City, Taiwan) with extended depth of field and cross-polarization. The digital microscope captures 5 megapixel (2952 × 1944) color images.

### 2.2. Design and Fabrication of the Dual Proximal Reflectance and Transillumination SWIR Probe

The dual proximal reflectance and transillumination SWIR probe was designed in Fusion 360 from Autodesk (San Francisco, CA, USA). The dual probe design consists of a handpiece with a reflectance and transillumination attachment, shown in Figure 2A–E. The handpiece was fabricated using a Formlabs Form 3 Low Force Stereolithography 3D printer. The final design was exported as an STL file and transferred to Formlabs PreForm to generate supports for a final 3D printing scheme. A spatial resolution of 100 µm was used for all prints. The probe consists of two components, the main body containing the light source for reflectance and the light collection optics and a second attachment containing the transillumination light source. The main body of the probe is very similar to the main body of our occlusal reflectance and transillumination probe with the exception that unpolarized light is used for reflectance [28]. The proximal transillumination attachment shown in Figure 2A consists of the main body of the probe containing the light source for reflectance and the light collection optics, a 3D-printed sliding frame (3), a black Delrin tube enclosing the transillumination light source (5), a rod holder (4) that connects (3) and (5), shown in Figure 2A. Broadband light from a tungsten halogen lamp with a bandpass filter at 1450 (with a bandwidth of 80 nm) is transmitted through a 1 mm diameter optical fiber (10) inserted into a 2 mm in diameter Teflon tube (8) enclosed in the probe’s back piece (2). There is an air nozzle positioned on the bottom of the probe opposite the light source for reflectance (2) directed towards the aluminum reflector to prevent fogging of the aluminum surface. The air nozzle can also be used to dry the tooth surface to increase lesion contrast and potentially assess lesion activity [31,32,33].

More details about the probe’s main body and reflectance backpiece (2) are described in Zhu et al. [28]. For proximal transillumination, the transillumination light source shown in Figure 2C is placed on the side of the tooth opposite to the main body of the probe as shown in Figure 2B. Several different proximal transillumination light diffusers were tested to determine which yields the highest contrast of interproximal lesions. Three different diffusing materials were tested: Teflon, Delrin, and Formlabs white resin, and several different diffuser geometries were investigated. The highest contrast was achieved using a 2 mm in diameter Teflon rod with a 400 µm optical fiber inserted in the center with 1300 nm light. The Teflon diffuser is shown in Figure 2B, and consists of a 400 µm optical fiber (11) connected to a superluminescent diode (SLD) enclosed inside the Teflon tube (9) that is placed inside a black Delrin tube (5). For maximum transillumination light intensity, a right-angled aluminum reflector (7) is fitted at the end of the Delrin rod to direct the diffuse light from the Teflon tube to the proximal contact between the teeth. The proximal transillumination attachment allows horizontal and vertical adjustments during sample screening so teeth with various shapes and sizes can be imaged with optimum light illumination, as shown by the black arrows in Figure 2A. The fully assembled in vitro imaging system is shown in Figure 2D. A version more suitable for clinical imaging was also designed utilizing Formlabs Flexible resin to connect the Delrin rod holder and the probe’s main body, and this is shown in Figure 2E. The flexible frame is composed of a soft and flexible material allowing clinicians to apply a squeezing force with the thumb to adjust light placement during clinical imaging.

### 2.3. Image Acquisition and Analysis

The SWIR reflectance images were captured using a Model GA1280J (Sensors Unlimited, Princeton, NK, USA) camera with a 1280 × 1024 pixel format, a 15 μm pixel pitch, and a bit depth of 12 bit. Two 1 inch in diameter planoconvex antireflection coated lenses of 60 mm and 100 mm focal length along with an adjustable aperture were placed between the handpiece and the InGaAs camera to provide a field of view of 11 mm × 11 mm at the focus plane. The transillumination light was delivered through a 0.4 mm diameter low-OH optical fiber. A 1312 nm superluminescent diode (SLD) from Covega (Jessip, MD, USA) with an output of 22 mW and a bandwidth of 50 nm was used as the light source for transillumination. The output intensity was set at 10 mw before entering the Teflon scattering rod. The reflectance and transillumination light sources were manually shut on and off and were not simultaneously acquired. The probe can also be used with the smaller 640 × 480 pixels micro-SWIR camera (SU640CSX) measuring only 32 mm × 32 mm × 28 mm from Sensors Unlimited (Princeton, NJ, USA) that is better suited for clinical imaging. The light for reflectance was delivered through a low-OH 1 mm optical fiber connected to a Model SLS202 extended wavelength tungsten-halogen light source from Thorlabs (Newton, NJ, USA) with a bandpass filter centered at 1460 nm with a bandwidth of 85 nm.

Images of the interproximal lesions were also acquired using a multispectral, dual occlusal transillumination and a reflectance imaging probe. A low-OH optical fiber of 1 mm diameter was used to deliver light from a 1604 nm superluminescent diode (SLD), Model ESL 1620–2111 from Exalos (Schlieren, Switzerland) with an output of 17 mW and a bandwidth of 46 nm. The intensity delivered to the tooth was 5 mW. The transillumination light was delivered through two 0.4 mm diameter low-OH optical fibers. A 1314 nm (BW) SLD, Model DL-CS3452A-FP from Denselight (Singapore) with an output of 48 mW and a bandwidth of 33 nm was used as the source for transillumination. A 50/50 beamsplitter was used to deliver light to each arm for transillumination. The output intensity of each arm was set at 10 mw before entering the Teflon plugs. Details regarding the fabrication of the dual occlusal transillumination and reflectance imaging probe system and optical probe have been previously described in [28].

The samples were dried of excess water with an air nozzle before imaging due to the strong water absorption at 1450 and 1600 nm [4]. Image processing of the images was performed by custom scripts written using MATLAB from Mathworks (Natick, MA, USA). The acquired 12-bit images (4096) were converted to 16-bit (65535) by multiplying by 16 and subtracting 1 to facilitate processing using MATLAB. Lesion areas were identified from the transillumination/reflectance images. Sound areas or regions of interest (ROI) for comparison were chosen from areas surrounding the lesion. This better represents the contrast between the lesion and sound tissues that would be viewed by the clinician. The contrast was calculated for each lesion using the formula (I_L_ − I_S_)/I_L_ for reflectance images and (I_S_ − I_L_)/I_S_ for transillumination images, where I_L_ is the average intensity in the lesion area and I_S_ is the average intensity in the sound ROI [5]. Lesion areas were confirmed using the microCT images.

The lesion depth and width in the XY plane and the distance to the occlusal surface (Z) were measured as shown in Figure 1 as appropriate for each imaging device. The lesion geometric information was measured in pixels first and then converted to micrometers imaging a reference ruler at the focal length for the SWIR images. It was not possible to perform a similar calibration for the radiographs. The linear correlation between the lesion dimensions measured with SWIR imaging methods and radiography with microCT was determined and correlation coefficients (R^2^) are reported if the correlation is significant (*p* < 0.05).

## 3. Results

MicroCT was used to image the 36 extracted teeth used for this study. There were 67 lesions visible on microCT images indicating that most of the teeth had lesions at both contact points (mesial and distal sides), i.e., 67 out of a possible 72 if lesions were present at all contact points, two per tooth. Out of the 67 lesions, 16 of the lesions were very small and were only visible with microCT. Only 22 of the 67 lesions had measurable contrast on dental radiographs for a detection rate of only 33%. Proximal transillumination at 1300 nm performed well showing measurable contrast for 40 out of 67 lesions, giving a 60% lesion detection rate. The lesion contrast was high for most proximal transillumination images, however, only seven of the lesions were visible in the proximal reflectance images of the buccal and lingual surfaces. There are two interproximal lesions on the tooth shown in Figure 3, one is deep and penetrates into the dentin while the second lesion is shallow and only penetrates halfway through the enamel. Both lesions are visible in the radiograph. Both lesions show significant contrast in the proximal transillumination mode. When imaged from the lingual side, Figure 3C, both lesions are visible, however only one of the lesions is visible when the imager was positioned on the buccal side of the tooth, Figure 3A. However, the two lesions are not visible in the corresponding reflectance images shown in Figure 3B,D, neither from the lingual nor from the buccal surface. Both lesions are clearly visible in the microCT image shown in Figure 3E.

Another tooth with two interproximal lesions is shown in Figure 4. One lesion penetrates almost to the dentin in the microCT image. The other lesion is very small and is barely visible in the microCT images. Occlusal transillumination and occlusal reflectance images are shown, and the larger lesion is visible in both the transillumination and reflectance images.

A major concern regarding the diagnosis of lesions on proximal surfaces is that radiographs markedly underestimate the depth of penetration [22,23,24]. The lesion penetration from the proximal surface (X_pt_) and distance to the occlusal surface (Z_pt_) were measured for the 40 lesions that were visible in the proximal transillumination mode images and compared with similar dimensions measured with microCT (X_µ_, Z_µ_) as shown in Figure 1. Similar measurements were not carried out for the proximal reflectance measurements since only seven of the lesions were visible in the proximal reflectance images. The lesion depth measured with proximal transillumination (X_pt_) and with microCT (X_µ_) is plotted in Figure 5 for the 40 lesions that were visible. There was a significant (*p* < 0.05) and moderate correlation between the two measurements with R^2^ equal to 0.40. There was also a significant correlation of the distance to the occlusal surface (Z_pt_) measured with proximal transillumination and with microCT (Z_µ_) but it was lower with R^2^ equal to 0.33, as shown in Figure 6. The teeth were also imaged using the other dual probe, the occlusal transillumination and reflectance probe and the sensitivity was higher, 51 out 67 of the lesions had measurable contrast in occlusal transillumination mode and 45/67 had contrast in occlusal reflectance mode for detection rates of 76 and 67% respectively. The depth of the interproximal lesions was also measured using occlusal transillumination (X_ot_) and reflectance (X_or_). Those measurements are plotted in Figure 7 against the depths measured with microCT (X_µ_). There was a significant (*p* < 0.05) and even higher correlation for occlusal transillumination (X_ot_) with R^2^ equal to 0.80. Lesion depth measurements with occlusal reflectance (X_or_) also correlated with microCT depth measurements (*p* < 0.05) but the correlation was not as high as for occlusal transillumination with R^2^ equal to 0.38. The lesion width was also measured with occlusal transillumination (Y_ot_) and occlusal reflectance (Y_or_). The lesion width measured with occlusal transillumination (Y_ot_) correlated with the lesion width measured with microCT (Y_µ_) with an R^2^ equal to 0.33. There was no correlation of the lesion width with microCT for the occlusal reflectance (Y_or_) images.

Lesion depths were also measured for radiography (X_r_) in pixels and those values are plotted against the microCT (X_µ_) measurements in Figure 8. There was no significant correlation (*p* > 0.05) of the measured lesion depths between radiography and microCT. The mean lesion contrast ± standard deviation, the correlation of the lesion depth with microCT, and the number of lesions that were visible in each SWIR imaging modality are tabulated in Table 1.

## 4. Discussion

In this study, a dual proximal transillumination and reflectance imaging device was fabricated and assessed for imaging interproximal lesions from the buccal/lingual tooth surfaces. The motivation for this device is to increase diagnostic performance by reducing false positives and providing more accurate measurements of lesion severity. The performance of proximal reflectance imaging from the buccal/lingual surface was disappointing, only 7 of the 67 interproximal lesions were visible. This result was not anticipated, prior clinical studies with occlusal reflectance measurements indicated that occlusal reflectance had a higher sensitivity than proximal and occlusal transillumination for the detection of caries lesions on both occlusal and proximal surfaces [9]. In this study, 45 of the 67 interproximal lesions were visible in occlusal reflectance images and the mean contrast of those lesions was higher than for the other SWIR imaging modes. Therefore, it appears that reflectance imaging is only effective for imaging interproximal lesions from the occlusal surface. The thickness of enamel between the tooth surface and interproximal lesions is similar for imaging from the tooth buccal/lingual and the occlusal surface. Therefore, the thickness of sound enamel between the tooth surface and the lesion does not explain the difference in performance. The most likely explanation is that the higher tooth curvature when imaging from tooth buccal and lingual surfaces increases refraction and total internal reflection and markedly reduces lesion contrast.

According to this study and three other studies, proximal and occlusal transillumination imaging and occlusal reflectance imaging performed as well or significantly better than radiography for the detection of lesions on proximal surfaces [2,9,28]. This study has also shown that proximal and occlusal transillumination imaging and occlusal reflectance imaging provide more accurate measurements of the depth and severity of the lesions compared to radiography. Accurate measurements of lesion severity are required for clinicians to make informed decisions about treatment. If the lesions are shallow and confined to the enamel, then chemical intervention with anti-caries and remineralization agents is warranted and the lesion can be followed over time to see if it continues to progress. If the lesion has penetrated and spread well into the dentin, then surgical intervention may be required. Since SWIR imaging methods do not involve exposure to ionizing radiation and they provide more accurate measurements of lesion depth penetration they are better suited for following the progression of lesions over time.

Proximal reflectance measurements from tooth buccal and lingual surfaces did not prove to be particularly useful for imaging interproximal lesions on tooth proximal surfaces, however, other types of lesions are often found on tooth buccal and lingual surfaces. Lesions are found in buccal and lingual pits and demineralization is very common around orthodontic brackets. Moreover, root caries and dental calculus on buccal and lingual surfaces are an increasing problem, and SWIR reflectance measurements at wavelengths greater than 1400 nm can image calculus and root caries with very high contrast [34]. Such surfaces are typically heavily stained and covered in plaque deposits, SWIR imaging is particularly valuable for differentiating demineralization and calcified plaque on those surfaces from stains and noncalcified plaque. SWIR reflectance imaging is also useful for assessing the severity of fluorosis on tooth buccal surfaces [35]. Therefore, even though the addition of proximal reflectance did not prove to be particularly valuable for imaging interproximal lesions on proximal surfaces, the dual proximal reflectance and transillumination device should be valuable for imaging other lesion types on tooth buccal and lingual surfaces.

Proximal transillumination was also useful for showing the distance of the lesion from the occlusal surface. That distance is also visible in radiographs although radiographs require exposure to ionizing radiation and the higher sensitivity of proximal transillumination indicates that many lesions will only be visible in proximal transillumination images and that the distance from the tooth surface will frequently not be accessible from a radiograph. In this study, almost twice as many lesions were visible in proximal transillumination compared to radiography (40 vs. 22). When clinicians restore teeth with interproximal lesions, they have to drill from the tooth occlusal surface, therefore knowledge of the depth of the lesion and its position is important.

Lesion widths in the XY plane were also examined and compared with microCT. The lesion width is not visible in radiographs however it can be measured from the occlusal surface using occlusal transillumination and reflectance. Lesion depths are of greater interest since they show the depth of penetration, however, the lesion width is also useful because it can be used to estimate the overall size of the lesion and is valuable for making a more informed decision on whether surgical intervention is recommended.

A major concern of SWIR/NIR imaging methods is the increased potential for false positives due to the higher sensitivity of these methods compared to radiography. A primary motivation for imaging the interproximal lesions using multiple SWIR imaging modalities is to reduce the potential for false positives. False positives may occur due to cracks, anatomical features in the tooth, and optical effects such as specular reflection. It is unlikely such effects would appear similarly in different imaging geometries and modalities. In order to be useful in this regard, the sensitivity needs to be high in multiple imaging modalities. We have shown that the sensitivity is high for proximal transillumination, occlusal transillumination, and occlusal reflectance imaging. The dual proximal transillumination reflectance device is not useful in this regard due to the low sensitivity of proximal reflectance imaging for interproximal lesions, however, the dual occlusal transillumination and reflectance device is well suited for reducing false positives due to the high sensitivity of both modes of imaging.

Different imaging appliances can be designed to attach to the SWIR imager and in a 2016 clinical study, three imaging probes were used to image premolar teeth scheduled for extraction; proximal transillumination and occlusal transillumination probes operating at 1300 nm and a reflectance probe at 1600 nm [9]. Another reason for the dual probe is to reduce the number of probes that have to be used and switched during clinical imaging. The two dual probes used in this study should be sufficient to image most of the lesions encountered and it is certainly feasible to design them so they can be rapidly switched in the clinic.

This study further demonstrates the value of multispectral transillumination and reflectance probes for imaging caries lesions. In this study, the first dual proximal transillumination and reflectance SWIR imaging device was assembled and the performance for imaging interproximal lesions on extracted teeth was assessed. The performance was compared with a dual occlusal transillumination and reflectance SWIR imaging device, radiography, and microCT. Proximal and occlusal transillumination at 1300 nm and occlusal reflectance measurements at 1600 nm provided significantly higher sensitivity than radiography for the detection of interproximal lesions and were more useful for measuring the lesion dimensions.

## 5. Conclusions

Occlusal and proximal transillumination and occlusal reflectance images were useful for imaging interproximal lesions and they provide more accurate measurements of lesion dimensions compared to radiography. Moreover, it has been shown that all SWIR imaging geometries relevant to imaging caries lesions can be implemented in two multispectral clinical probes, one that acquires occlusal transillumination and reflectance images and the other that acquires transillumination and reflectance images.

## Figures and Tables

**Figure 1 diagnostics-12-00597-f001:**
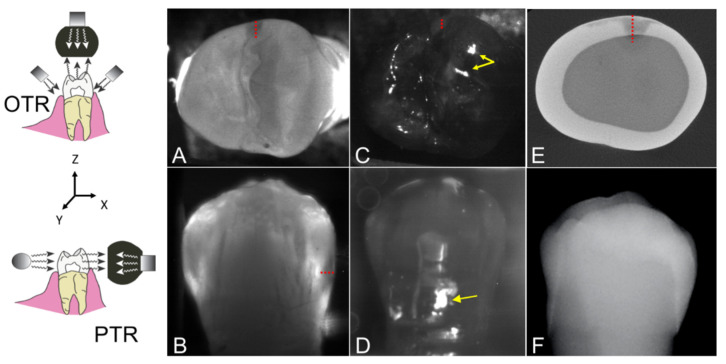
Diagrams of the occlusal (OTR) and proximal (PTR) dual short-wavelength infrared (SWIR) transillumination and reflectance probes used in this study. The orientation of the XYZ axes with respect to the long axis of the tooth is shown. (**A**) Occlusal transillumination image, (**B**) proximal transillumination image (buccal surface), (**C**) occlusal reflectance image and (**D**) proximal reflectance (buccal surface) image, (**E**) transverse microCT slice in the XY plane of a tooth, and (**F**) dental radiograph of a tooth with an interproximal lesion is shown. The red dotted line represents the lesion depth from the proximal surface (X direction), the green dotted line represents the lesion width (Y direction) and the blue dotted line represents the vertical distance of the lesion to the tooth occlusal surface (Z direction). The yellow arrows in the reflectance images (**C**,**D**) are areas of specular reflection that can potentially cause false positives.

**Figure 2 diagnostics-12-00597-f002:**
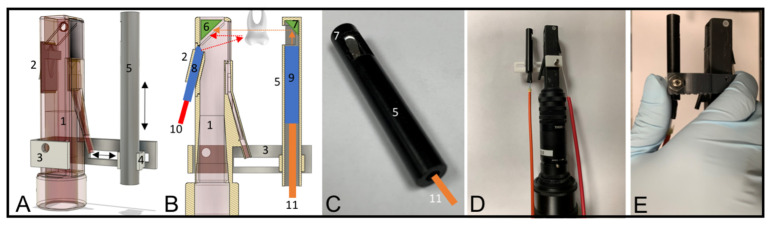
(**A**) 3D rendering and (**B**) cross-section of the dual SWIR proximal probe is shown. Major parts include: (1) probe main body, (2) reflectance backpiece, (3) sliding frame, (4) Delrin rod holder, (5) Delrin tube, (6), right-angled rectangular aluminum reflector, (7) right-angle cylindrical aluminum reflector, (8) Teflon tube for reflectance light source, (9) Teflon tube for proximal transillumination light source, (10) optical fiber delivering 1450 nm light for reflectance, (11) optical fiber delivering 1300 nm light for proximal transillumination. Proximal transillumination light propagation is shown in orange dashed arrows. Reflectance light propagation is shown with red dashed arrows. (**C**) Proximal transillumination light source (11) is encased in a Teflon tube placed inside a Delrin tube (5). The cylindrical aluminum mirror (7) reflects anisotropically scattered light from the Teflon tube and directs it to the sample. (**D**) Fully assembled dual SWIR proximal probe for in vitro use. (**E**) For in vivo use, the sliding frame (3) can be replaced with a soft/flexible holder.

**Figure 3 diagnostics-12-00597-f003:**
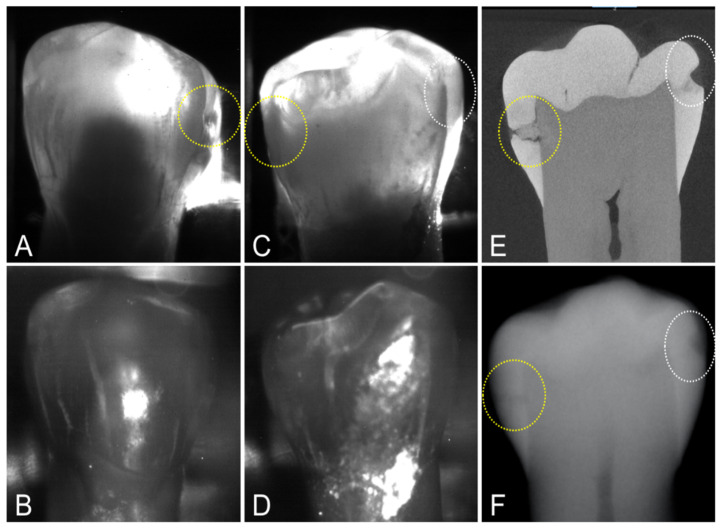
Proximal transillumination and reflectance images of a tooth with two interproximal lesions on opposing proximal surfaces. Lesion 1 (yellow circle) is more severe and penetrates to the dentin while lesion 2 (white circle) is limited to the enamel. (**A**) Transillumination and (**B**) Reflectance images with the imager on the buccal side of the tooth. (**C**) Transillumination and (**D**) Reflectance with the imager on the lingual side of the tooth. (**E**) MicroCT slice (**F**) and Dental X-Radiograph are also shown. Neither lesion is visible in the proximal reflectance images.

**Figure 4 diagnostics-12-00597-f004:**
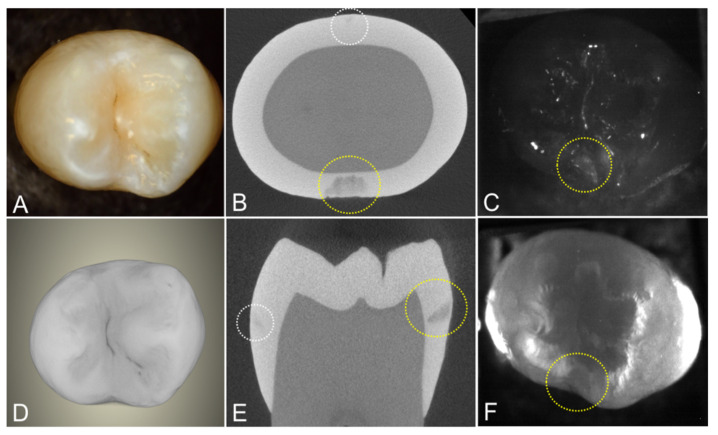
Images of a tooth with two interproximal lesions, one that is quite small and is only visible in the microCT images (white circle) and one much larger that was also visible in the SWIR images (yellow circle). (**A**) color and SWIR (**C**) occlusal reflectance and (**F**) occlusal transillumination are shown along with a (**D**) microCT surface rendering of the tooth and extracted slices (**B**) transverse and parallel. (**E**) to the long axis of the tooth.

**Figure 5 diagnostics-12-00597-f005:**
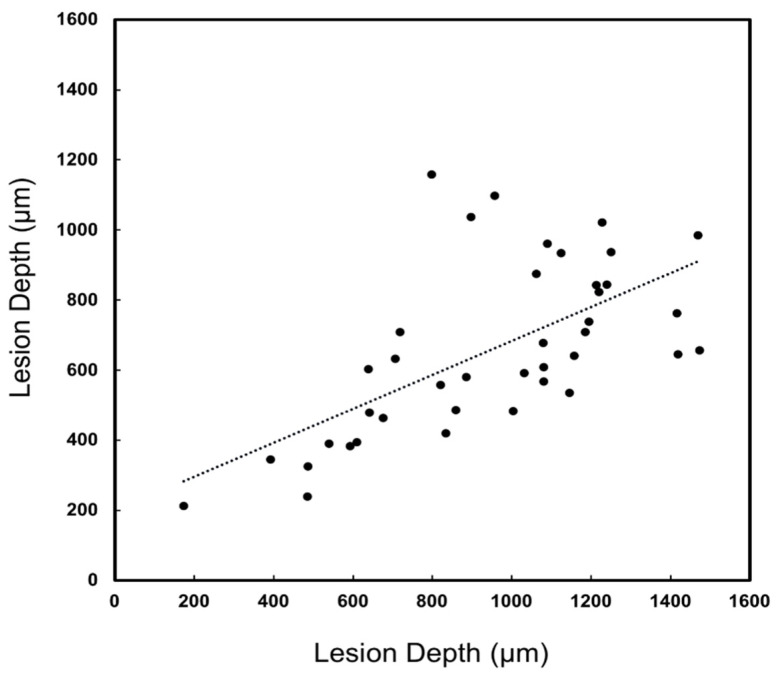
The lesion depth was measured in proximal transillumination mode (X_pt_) and microCT (X_µ_). There was a significant correlation between the two methods (*p* < 0.05) with R^2^ = 0.40 and 40 of the 67 lesions had sufficient contrast to perform the measurements.

**Figure 6 diagnostics-12-00597-f006:**
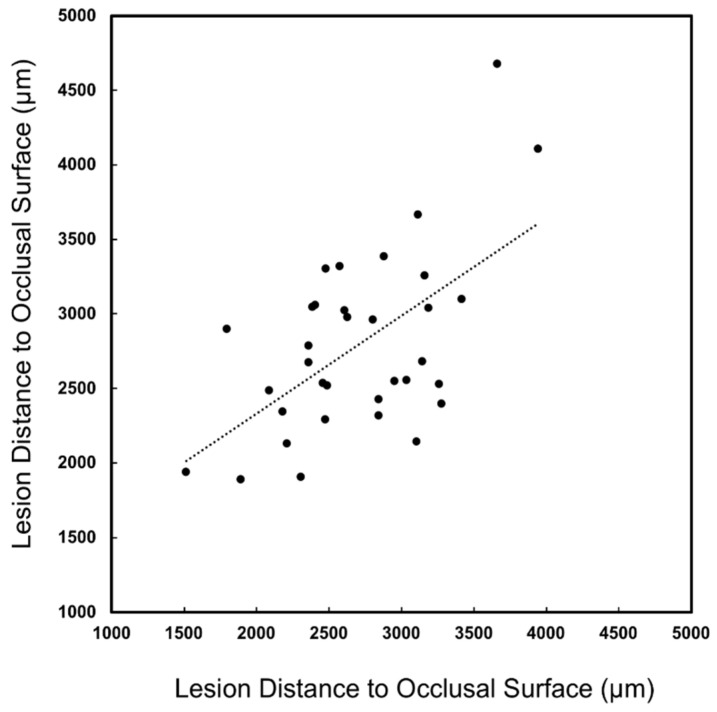
Depth below the occlusal surface measured in transillumination mode (Z_ot_) and microCT (Z_µ_) There was a significant correlation between the two methods (*p* < 0.05) with R^2^ = 0.33 and 34 of the 67 lesions had sufficient contrast to perform the measurements.

**Figure 7 diagnostics-12-00597-f007:**
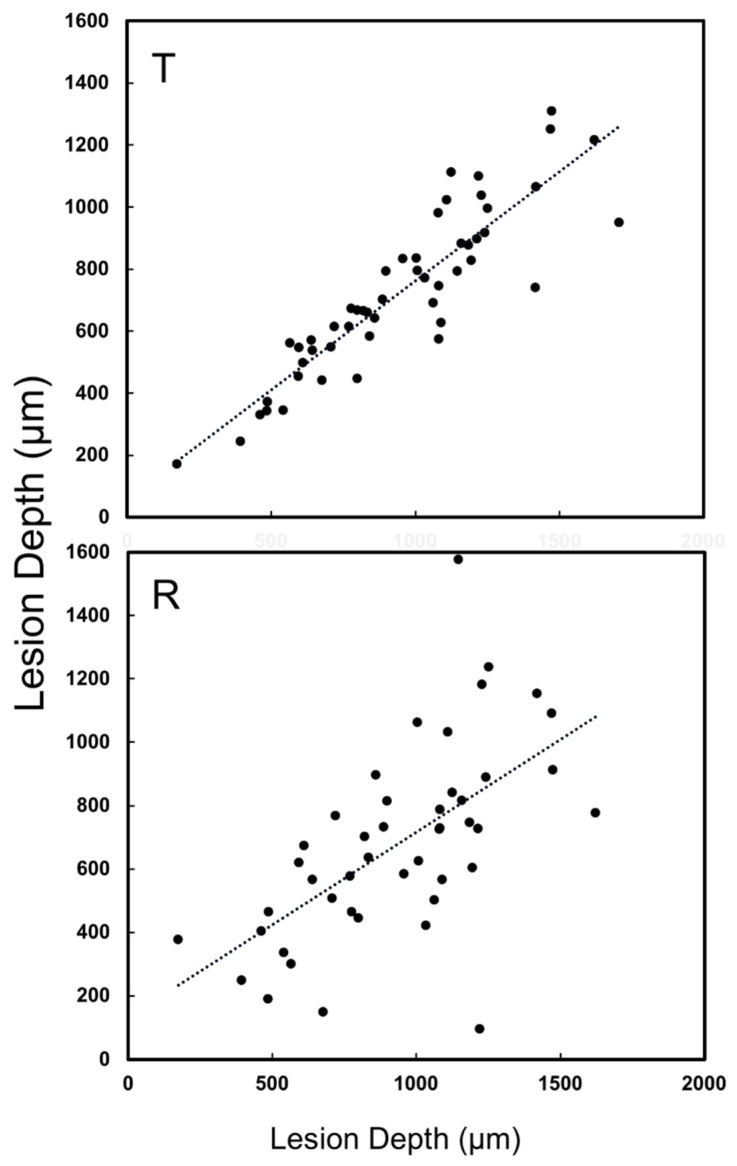
(T) Lesion depth measured in occlusal transillumination mode (X_ot_) and microCT (X_µ_). There was a significant correlation between the two methods (*p* < 0.05) with R^2^ = 0.80 and 51 out of the 67 lesions had sufficient contrast to perform the measurements. (R) Lesion depth measured in occlusal reflectance mode (X_or_) and microCT (X_µ_). There was a significant correlation between the two methods (*p* < 0.05) with R^2^ = 0.38 and 45 of the 67 lesions had sufficient contrast to perform the measurements.

**Figure 8 diagnostics-12-00597-f008:**
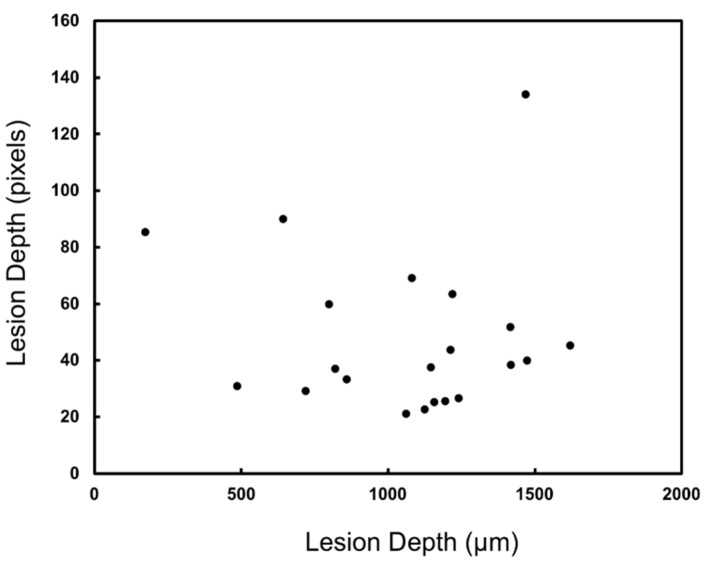
Lesion depth was measured in radiography (X_r_) and microCT (X_µ_). There was no significant correlation between the two methods (*p* > 0.05). Only 22 of the 67 lesions had sufficient contrast to perform the measurements.

**Table 1 diagnostics-12-00597-t001:** Mean lesion contrast ± s.d. for the SWIR methods and radiography and linear correlation coefficients (R^2^) for the lesion depth measured with those methods compared with microCT. The number of lesions that had measurable contrast in each imaging modality out of the 67 lesions identified with microCT is also listed.

	Proximal Transillumination	Occlusal Transillumination	Occlusal Reflectance	Radiograph
Mean Lesion Contrast ± s.d.	0.49 ± 0.11	0.45 ± 0.16	0.55 ± 0.21	0.26 ± 0.10
Lesion Depth Correlation (R^2^)	0.40	0.80	0.38	N/A
Lesions Visible (N)	40	51	45	22

## Data Availability

Not applicable.

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
