# Peer review of "Measurement of the Depth of Lesions on Proximal Surfaces with SWIR Multispectral Transillumination and Reflectance Imaging"

_diagnostics, 2022, doi:10.3390/diagnostics12030597_

Round 1
Reviewer 1 Report
Dear authors, Congratulations! You have successfully added a further piece into the big mosaic. The terminology "Caries Diagnosis" is surely both adequate and confusing at the same time. The "SWIR Multispectral Transillumination and Reflectance Imaging" empowers the identification of cavitated lesions. The reviewer would suggest a more precise definition for the described approach to help the reader better understand the purpose of the procedure from start. This is a great and valuable piece of research deserving immediate publication! Thank you for submitting your work to this journal!
Author Response
We greatly appreciate the kind words and the efforts of the reviewer to improve the manuscript and we hope we have sufficiently addressed their concerns.
Changed title to :
Measurement of the Depth of Lesions on Proximal Surfaces with SWIR Multispectral Transillumination and Reflectance Imaging
Reviewer 2 Report
I thank the Editorial Office for giving me the opportunity to review this interesting article and I would like to congratulate the authors for the good work they have done: the article is very well articulated and described in detail.
I just have a few small comments, which I hope will help the authors make their article easier to understand.
The authors mention figure number 2 before number 1, creating some confusion in the reader who does not understand if the error is in the text or in the order of the photo captions.
It would be more correct to provide a bibliographic citation to the sentences "Radiographs markedly underestimate the depth and severity of interproximal lesions" (lines 69-70) and “radiographs markedly underestimate the depth of penetration” (241-243).
What do the authors mean by "surgical intervention" (lines 72 and 74) since these procedure do not imply surgical procedures?
The authors state that "According to this study and two other studies, proximal and occlusal transillumination imaging and occlusal reflectance imaging performed as well or significantly better than radiography for the detection of lesions on proximal surfaces [2,9,25]" but they cite three work instead of two. Please correct the text or bibliographic entries.
Author Response
We greatly appreciate the kind words and the efforts of the reviewer to improve the manuscript and we hope we have sufficiently addressed their concerns.
The authors mention figure number 2 before number 1, creating some confusion in the reader who does not understand if the error is in the text or in the order of the photo captions.
We have switched the position of the two figures so that figure number 1 appears first in the text.
It would be more correct to provide a bibliographic citation to the sentences "Radiographs markedly underestimate the depth and severity of interproximal lesions" (lines 69-70) and “radiographs markedly underestimate the depth of penetration” (241-243).
We have added three references (22-24) referring to the accuracy of proximal caries depth to both of those sentences.
What do the authors mean by "surgical intervention" (lines 72 and 74) since these procedure do not imply surgical procedures?
Replaced with:
If lesions are noncavitated and limited in penetration to only the enamel they can often be arrested by chemical intervention without the need to remove the decay with the drill. Therefore, a more accurate imaging system for assessing the depth of penetration of interproximal lesions would be a significant step forward and enable clinicians to make more informed decisions regarding treatment.